genetics/molecular biology

microsatellites (SSR), tetraploid, gene flow, genetic differentiation, conservation

**Author for correspondence:**
Bohdan Lojka
e-mail: lojka@ftz.czu.cz

# Genetic diversity and structure of baobab (*Adansonia digitata* L.) in southeastern Kenya

Anna Chládová[1], Marie Kalousová[1], Bohumil Mandák[2,3], Katja Kehlenbeck[4,5], Kathleen Prinz[6], Jan Šmíd[2], Patrick Van Damme[1,4,7] and Bohdan Lojka[1]

[1]Department of Crop Sciences and Agroforestry, Faculty of Tropical AgriSciences, and
[2]Department of Ecology, Faculty of Environmental Sciences, Czech University of Life Sciences Prague, Kamýcká 129, 165 00 Prague, Czech Republic
[3]The Czech Academy of Sciences, Institute of Botany, Zámek 1, 252 43 Průhonice, Czech Republic
[4]World Agroforestry Centre (ICRAF), PO Box 30677, 00100 Nairobi, Kenya
[5]Rhine-Waal University of Applied Sciences, Marie-Curie-Straße 1, 47533 Kleve, Germany
[6]Institute for Systematic Botany, Friedrich-Schiller-University Jena, Philosophenweg 16, 07743 Jena, Germany
[7]Department of Plants and Crops, Faculty of Bioscience Engineering, Ghent University, 9000 Ghent, Belgium

AC, 0000-0001-8427-062X; MK, 0000-0001-7078-4064;
BM, 0000-0002-9545-7497; PVD, 0000-0002-2548-633X;
BL, 0000-0002-5059-1975

Baobab (*Adansonia digitata* L.) is an iconic tree of African savannahs. Its multipurpose character and nutritional composition of fruits and leaves offer high economic and social potential for local communities. There is an urgent need to characterize the genetic diversity of the Kenyan baobab populations in order to facilitate further conservation and domestication programmes. This study aims at documenting the genetic diversity and structure of baobab populations in southeastern Kenya. Leaf or bark samples were collected from 189 baobab trees in seven populations distributed in two geographical groups, i.e. four inland and three coastal populations. Nine microsatellite loci were used to assess genetic diversity. Overall, genetic diversity of the species was high and similarly distributed over the populations. Bayesian clustering and principal coordinate analysis congruently divided the populations into two distinct clusters, suggesting significant differences between inland and coastal populations. The genetic differentiation between coastal and inland populations suggests a limited possibility of gene flow between these populations. Further conservation and domestications studies should take into consideration the geographical origin of trees and more attention should be paid to morphological characterization of fruits and

leaves of the coastal and inland populations to understand the causes and the impact of the differentiation.

# 1. Background

The multipurpose tree species *Adansonia digitata* L., well-known as 'baobab', is one of the most important indigenous fruit trees of sub-Saharan Africa, with substantial social and economic importance [1,2]. The tree provides food, medicine and fibre, whereas almost all tree parts, i.e. fruit pulp and shells, seeds, leaves, flowers, roots and bark can be used [3]. Baobabs contribute significantly to economy of many rural communities [4]. Mainly leaves and fruits are collected for food and sold as raw or processed into a variety of products at local or international markets.

Baobab is considered to be a semi-domesticated species [5]. Humans have played an important role in current baobab distribution and the species' genetic structure. Local villagers, farmers and traders have been transporting fruits from village to village, and therefore contributing significantly to baobab dispersal and gene flow [5], not only limited to Africa but also to other continents, e.g. Caribbean Islands and India [6].

However, with increasing market demand and changing climate, the pressure on natural populations may negatively influence their viability. Domestication and sustainable cultivation of baobabs can be an option to increase production and at the same time conserve current natural populations [7]. However, to achieve sustainable domestication, characterization of population genetics is an important step. Such findings provide valuable insights into the genetic diversity, history of the species, population structure and therefore help with further tree management, breeding and conservation strategies. Results of this study may help to select elite baobab trees for domestication from distinct genetic pools, thus minimizing the loss of genetic diversity that often happens during domestication [8].

In the current study, the genetic characterization of baobab populations from Kenya is assessed by microsatellites (single sequence repeat, SSR) markers. Microsatellites are molecular markers suitable for various applications such as parentage analysis, conservation genetics, phylogeography, population genetics, etc. (as cited in [9]). Even though newer molecular methods are available, such as genotyping by sequencing, or restriction site-associated DNA sequencing, microsatellites are still effective and cost-efficient markers to use [9]. However, owing to the fact that baobab is tetraploid, the analysis of genetic data and their interpretation requires more scrutiny than with diploid species [10]. Studies on baobab genetic resources have been reported from West African countries such as Benin, Ghana, Burkina Faso and Senegal [5,11,12], while East Africa is represented by studies from Malawi and Sudan [2,13]. In studies from Sudan and Malawi, microsatellite loci developed for *A. digitata* [14] were applied, while studies from West Africa used amplified fragment length polymorphism (AFLP) markers. In an extensive genetic study of chloroplast DNA, analysing samples across the whole African continent [15], the authors found clear differentiation of the populations of *A. digitata* of West African versus South/East African origin. From East Africa, *A. digitata* was introduced by migrating people to the Indian subcontinent, most probably from coastal regions of Kenya and Tanzania [6]; however up until now, not much is known about genetic structure within East African countries, e.g. among coastal and inland populations.

In Kenya, baobab can be found in the eastern part of the country in two belts; the inland belt (from the Tanzanian border east of Mount Kilimanjaro towards the northeast around Kitui Town) and coastal belt [16]. In Kenya, it occurs in a wide altitudinal range (from sea level up to 1058 m above sea level (m.a.s.l)) and in diverse vegetation types such as deciduous bushland, woodland and wooden grassland, but also thrives in areas with high rainfall on the coast [16,17]. These broad and diverse occurrences of baobabs in Kenya are not yet thoroughly studied from the genetic point of view. Therefore, the main objective of the study was to assess the level of genetic diversity and the genetic structure of baobab populations in southeastern Kenya using highly variable microsatellite markers. The results of this study will contribute to the general knowledge of baobab genetic resources in East Africa and will be useful in further conservation and domestication programmes.

# 2. Material and methods

## 2.1. Study site

The present study was carried out in the Makueni and Taita Taveta Counties located in inland southeastern Kenya, and in Kwale and Kilifi Counties located on the Kenyan coast. The inland

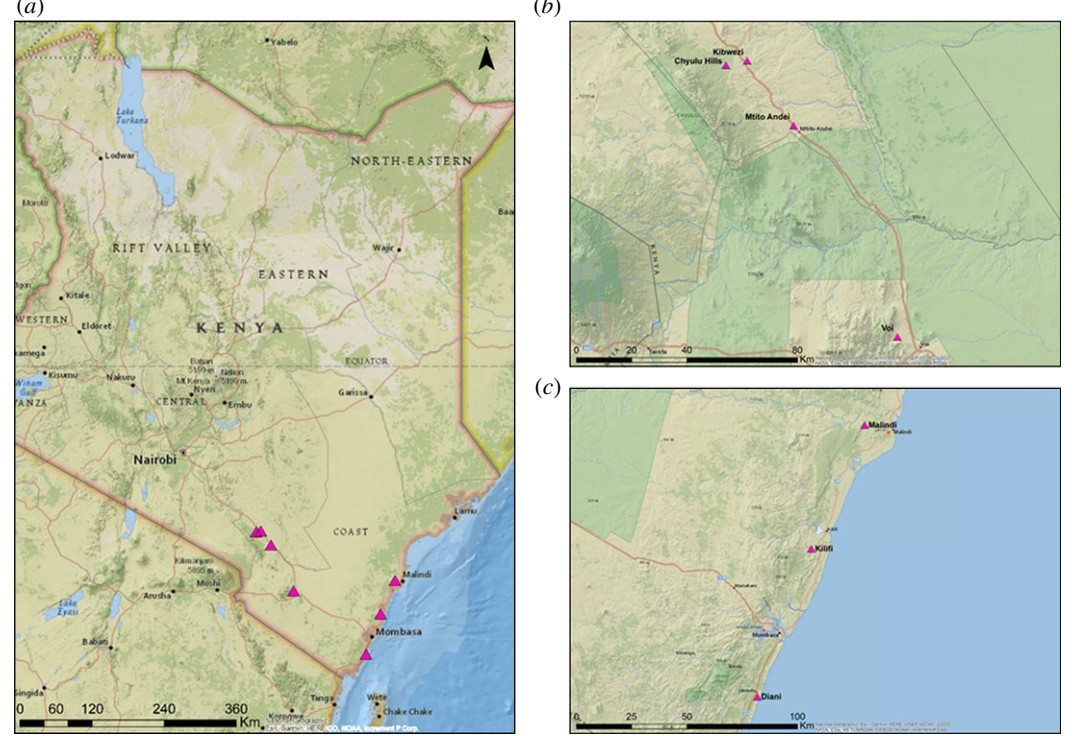

**Figure 1.** Map of Kenya, displaying the sampled baobab populations. (*a*) An overview map of all populations, (*b*) inland populations: Kibwezi, Mtito Andei, Chyulu, Voi, and (*c*) coastal populations: Diani, Malindi and Kilifi.

locations have a rather hot, semi-arid climate with an average rainfall from 500 to 700 mm yr$^{-1}$, a mean annual temperature of 23°C, and the most common soils are Luvisols and Ferralsols [18]. Trees from inland populations were sampled at an altitude range from 657 to 1058 m.a.s.l. The coastal sampling locations occur in tropical savannah climate with the mean annual rainfall ranging from 500 to 1500 mm, mean annual temperature of 26°C, and soils belong to many different types such as Luvisols, Ferralsols, Arenosols, Planosols, Vertisols or Acrisols [19]. Trees sampled in coastal areas were found at low altitude ranging from 6 to 54 m.a.s.l.

## 2.2. Tree sampling and data collection

In total, 189 individual trees were sampled from seven populations (inland populations: Kibwezi, Mtito Andei, Chyulu, Voi, coastal populations: Diani, Malindi, and Kilifi) during July–August 2015, figure 1 and table 1. The sampling was designed along transects corresponding to the Nairobi-Mombasa-Highway and along the coastline to cover the broad altitudinal range (from 6 to 1058 m.a.s.l.) of baobab occurrence in Kenya. The trees were sampled on 64 pre-selected farms along the transect ($n = 146$) (K. Kehlenbeck & C. Waruhiu 2015, unpublished data), and in the Chyulu Hills National Park ($n = 43$), a population which provided baobab trees at the highest elevation in the sampling area.

For each of the sampled trees, one to two healthy leaves were collected or about 5 cm$^2$ bark sample as an alternative source of DNA if no leaves were present. Samples were inserted into permeable paper bags, labelled, stored in sealed plastic boxes with silica gel for drying and transported to Prague, Czech Republic, for genetic analysis at the molecular laboratory of Czech University of Life Sciences Prague (CULS).

## 2.3. DNA extraction and SSR analyses

For DNA extraction, the cetrimonium bromide (CTAB) method was used [22]. Plant material was ground and powder was subsequently suspended in a CTAB lysis buffer (2% CTAB, 1.4 mol l$^{-1}$ NaCl, 0.1 mol l$^{-1}$ Tris–HCl pH 8.0, 20 mmol l$^{-1}$ ethylenediaminetetraacetic acid (EDTA), 1% v/v polyvinylpyrrolidone, 0.2% v/v 2-mercaptoethanol) for 30 min at 60°C. The suspension was incubated with an equal volume of chloroform-isoamylalcohol (24 : 1), and DNA was precipitated from the aqueous phase by 0.7 volumes of isopropanol. After washing with 70% ethanol, DNA was dissolved in water and treated by RNase A

**Table 1.** Location of collected baobab (*A. digitata*) populations and summary of genetic parameters within seven populations, based on nine microsatellite loci [14]. (The table contains mean values for each population, and mean values for inland and coastal accessions. Values for NA, NAe, He, Ho, $F_{is}$ were computed in SPAGeDi [20]. Nei's genetic diversity (*H*) was computed from binary data in GenAlEx 6.5 [21]. Latitude and longitude for each population were calculated as central points; altitudes are presented as mean values.)

| population | population characterization | latitude | longitude | altitude (m) | number of individuals n | Nei's genetic diversity H | observed number of alleles NA | effective number of alleles NAe | gene diversity (corrected for sample size) He | observed heterozygosity Ho | individual inbreeding coefficient $F_{is}$ |
|---|---|---|---|---|---|---|---|---|---|---|---|
| Kibwezi | inland | −2.458961 | 37.994513 | 926 | 78 | 0.153 | 11 | 5.22 | 0.788 | 0.913 | −0.120 |
| Mtito Andei | inland | −2.670773 | 38.145645 | 785 | 19 | 0.156 | 9 | 5.70 | 0.795 | 0.933 | −0.146 |
| Voi | inland | −3.360530 | 38.483770 | 709 | 23 | 0.144 | 9 | 4.73 | 0.770 | 0.911 | −0.141 |
| Chyulu Hills | inland | −2.473330 | 37.924853 | 1003 | 43 | 0.163 | 11 | 6.00 | 0.811 | 0.938 | −0.135 |
| Diani | coastal | −4.308418 | 39.568872 | 14 | 9 | 0.116 | 7 | 3.99 | 0.712 | 0.788 | −0.015 |
| Kilifi | coastal | −3.704866 | 39.788302 | 41 | 11 | 0.124 | 6 | 4.20 | 0.721 | 0.836 | −0.078 |
| Malindi | coastal | −3.199580 | 40.005385 | 26 | 6 | 0.124 | 5 | 4.79 | 0.753 | 0.812 | 0.049 |
| all populations | | | | | 189 | 0.140 | 13 | 5.72 | 0.803 | 0.907 | −0.085 |
| inland | | | | | 163 | 0.154 | 10 | 5.41 | 0.791 | 0.924 | −0.136 |
| coastal | | | | | 26 | 0.121 | 6 | 4.33 | 0.729 | 0.812 | −0.015 |

($250\ \mu g\ ml^{-1}$). DNA concentration and purity were measured on a NanoDrop™ spectrophotometer. Samples were diluted to final concentrations of $25\ ng\ \mu l^{-1}$ for further polymerase chain reaction (PCR).

Nine microsatellite primer pairs (Ad01, Ad05, Ad06, Ad08, Ad09, Ad11, Ad13, Ad17, Ad18) originally developed for *A. digitata* by Larsen *et al.* [14] were applied to the 189 samples. Forward primers were fluorescently labelled for detection by four colours: 6-FAM (Generi Biotech, Czech Republic), PET, NED and VIC (Thermo Fisher Scientific, USA). Two multiplex PCRs were established: multiplex PCR 1 consisted of six primer pairs: Ad05, Ad08, Ad09, Ad13, Ad17, Ad18; multiplex PCR 2 contained three primer pairs: Ad01, Ad06, Ad11. The PCR reaction mixture was set up in a total volume of $12\ \mu l$ containing $1\ \mu l$ of DNA ($25\ ng\ \mu l^{-1}$), $4.4\ \mu l$ of primer mix (each primer 10 mM), $0.9\ \mu l$ of 10× TaqBuffer (Thermo Fisher Scientific), $0.9\ \mu l$ of 10 mM dNTP Mix (Thermo Fisher Scientific), $0.9\ \mu l$ of 2.5 mM $MgCl_2$ (Thermo Fisher Scientific), $0.7\ \mu l$ 5 U of *Taq* DNA polymerase (Thermo Fisher Scientific) and $3.2\ \mu l$ of nuclease-free water. PCR amplifications were conducted separately for primer mix 1 (PM1) and primer mix 2 (PM2) in a T100 Thermal Cycler (Bio-Rad, USA) by performing the following steps: 2 min of denaturation at 95°C followed by 36 cycles at 95°C for 30 s, 58°C for 1 min and elongation at 72°C for 2 min 30 s and a final extension for 7 min at 72°C. PCR products of PM1 and PM2 were mixed in a ratio of 1 : 2 to create a multiplex for fragment analysis by capillary electrophoresis. The PCR products thus prepared were mixed with GeneTrace 500 LIZ size standard (Carolina Biosystems, Czech Republic) and analysed on a Genetic Analyser 3500 (Applied Biosystems, USA). Microsatellite alleles were scored using GeneMarker v. 2.4.0 (SoftGenetics, USA).

## 2.4. Data analysis

### 2.4.1. Genetic diversity and population differentiation

Considering the tetraploid nature of *A. digitata*, we used software SPAGeDi v. 1.5 that can handle polyploid species [20]. The number of alleles (NA), effective number of alleles (NAe, [23]), gene diversity corrected for sample sizes (He, [24]), observed heterozygosity (Ho) and inbreeding coefficients ($F_{is}$) were computed for all populations. The same program was used to estimate the pairwise population differentiation $\rho$ statistic [25] and geographical differentiation among populations. According to Meirmans *et al.* [10], the $\rho$ statistic is the most relevant alternative to $F_{ST}$ to assess the population differentiation for polyploid species. Microsatellite data from tetraploid *A. digitata* were converted to a binary matrix to calculate Nei's genetic diversity ($H$) and analyses of molecular variance (AMOVA). $H$ was computed from binary data in GenAlEx 6.5 [21], and the binary matrix was also applied for an AMOVA by Arlequin 3.5 [26]. Two different approaches, i.e. a direct estimation of allele frequencies based on SPAGeDi and a conversion of allele data to a binary presence/absence matrix, were applied to obtain sufficient results.

### 2.4.2. Population structure

Owing to allele copy ambiguity in tetraploid datasets, the determination of precise genotypes and allele frequencies was impossible. Here, the coding of microsatellite data was employed using the Polysat package [27] in R 3.1.1 [28]. Using this package, a principal coordinate analysis (PCoA) was performed based on the matrix of Bruvo distances between individuals. The input files to software SPAGeDi and Structure 2.3.4 [29] were prepared using Polysat. Bayesian model-based clustering of microsatellite data was computed in Structure 2.3.4.

To reveal the number of clusters, samples were divided into three datasets: one containing all samples, one with only inland samples and one with only coastal samples. Janes *et al.* [30] reported that $\Delta K$ frequently identifies $K = 2$ even if more subpopulations are present; therefore, it may cause an over- or underestimation of population genetic structure. To prevent this, a division of samples was done to reveal if there is more genetic structure within each of the two regions. In Structure, the admixture model and correlated allele frequency model were applied [29]. Then, Structure was run for each dataset with a burn-in period of 100 000 followed by 100 000 Markov chain Monte Carlo steps. For each dataset, the number of clusters ($K$) was adjusted as follows: $K = 1–10$ for all samples, $K = 1–7$ for inland samples and $K = 1–6$ for coastal samples. Each $K$-value was replicated 10 times. Owing to the unbalanced sampling of populations, the ALPHA ($\alpha$) parameters were set as follows: for all individuals $\alpha = 0.1$, for inland samples $\alpha = 0.14$, and for coastal samples $\alpha = 0.17$ [31]. The output data from Structure were analysed in the Structure Harvester [32] to assess the optimal $K$ following

**Table 2.** Pairwise geographical (above diagonal) and genetic differentiation ($\rho$, [25]) among baobab (*A. digitata*) populations from the seven studied locations (below diagonal) in southeastern Kenya.

| | Kibwezi | Mtito Andei | Voi | Chyulu Hills | Diani | Kilifi | Malindi |
|---|---|---|---|---|---|---|---|
| Kibwezi | | 29 | 114 | 8 | 270 | 243 | 238 |
| Mtito Andei | 0.000 | | 85 | 33 | 241 | 216 | 215 |
| Voi | 0.022 | 0.040 | | 116 | 160 | 150 | 170 |
| Chyulu Hills | 0.004 | 0.000 | 0.035 | | 274 | 248 | 245 |
| Diani | 0.289 | 0.313 | 0.330 | 0.289 | | 71 | 132 |
| Kilifi | 0.288 | 0.302 | 0.320 | 0.291 | 0.044 | | 61 |
| Malindi | 0.296 | 0.306 | 0.331 | 0.305 | 0.148 | 0.071 | |

the $\Delta K$ method of Evanno *et al.* [33]. Cluster alignment of replicates was conducted in CLUMPP v. 1.1.2 [34], and graphically visualized with DISTRUCT v. 1.1 [35].

# 3. Results

## 3.1. Genetic diversity

A total of 117 alleles were scored among nine loci. Allele frequencies and gene diversity computed per population are shown in table 1. The effective number of alleles per population ranged from 4 to 6, while the number of alleles per locus ranged from 5 to 11. Overall gene diversity (He) was high for all populations (greater than 0.71) (table 1). The overall observed heterozygosity (Ho) was also high (greater than 0.79); the Ho for single populations was always higher than He. The mean values of the inbreeding coefficient ($F_{is}$) ranged from –0.136 to 0.049 (table 1), the negative values indicate a slight deficit of homozygotes, the values closer to zero exhibit rather a balance between homozygotes and heterozygotes within populations. Similar values were found for combined inland and coastal individuals, with lower $F_{is}$ in inland populations resulting from higher observed heterozygosity.

Genetic differentiation ($\rho$) computed for the seven populations and compared with the geographical distances are presented in table 2. While genetic differentiation ($\rho$) between coastal and inland populations was generally high (greater than 0.090), differentiation within each of the two regions was very low. The lowest genetic differentiation was revealed between Mtito Andei and Kibwezi (0.0) as well as Chyulu populations (0.004). These were also the locations with short geographical distances of only about 29–33 km each. The highest values of genetic differentiation were found between Voi and Diani as well as Voi and Malindi (range 0.330–0.331) (table 2). In general, $\rho$ values among the four studied inland populations were low, ranging from 0.000 to 0.040, but moderate among the three studied coastal populations (0.044–0.148).

AMOVA calculated for all seven populations revealed low variation among (9.4%) and high variation within populations (90.6%). Separating inland and coastal regions, data showed 22.1% variance among and 77.9% within groups. AMOVAs conducted separately for inland and coastal populations revealed low variation among populations (table 3).

## 3.2. Population structure

The true number of clusters among all samples was detected for $K = 2$ (figure 2*a*), based on the Evanno *et al.* [33] statistics obtained from STRUCTURE HARVESTER [32]. These results are presented in the electronic supplementary material, S2. The bar plot resulting for two clusters clearly differentiated inland and coastal individuals. To reveal if there is a genetic structure within inland and coastal populations, subclustering analyses were performed for each region. For inland populations, we determined optimal $K = 4$ and for coastal populations $K = 3$. However, we did not identify any clear structure within either of the two regions (figure 2*b*,*c*). Similar results to the STRUCTURE analysis were obtained from the PCoA (figure 3). Within the inland and coastal clusters, individuals did not reflect any population structure but were clearly separated by the region.

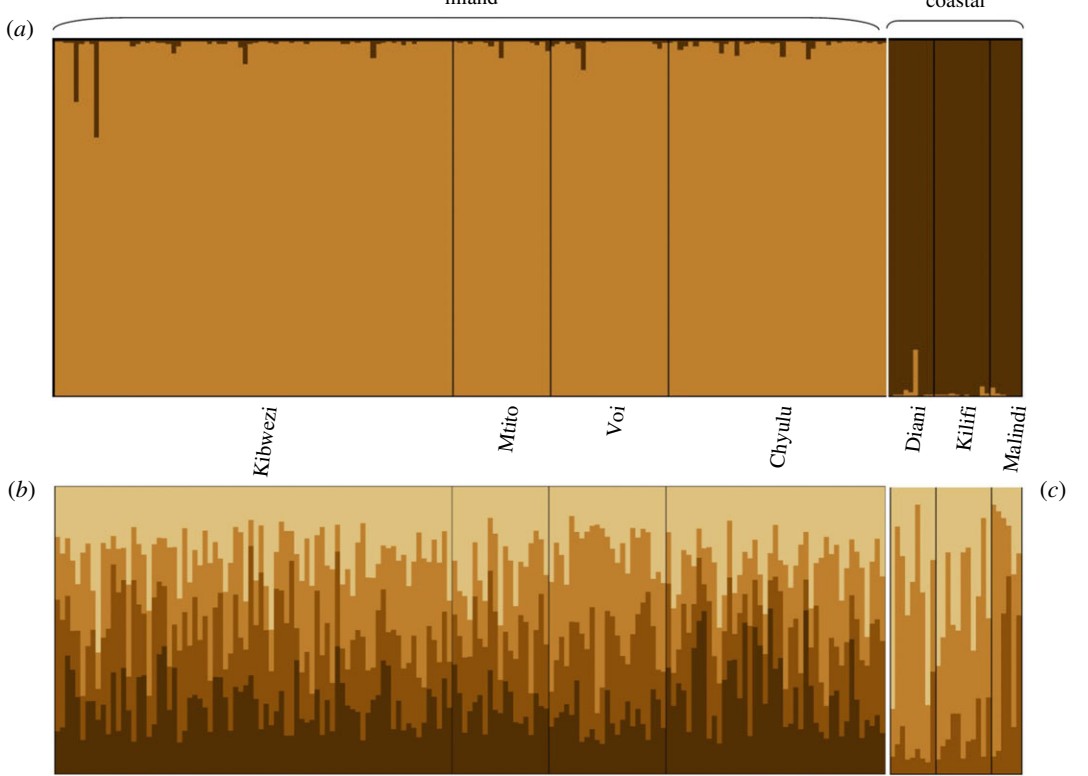

**Figure 2.** Bayesian clustering analysis based on nine SSR markers data for baobab (*A. digitata*) populations from southeastern Kenya performed using Sᴛʀᴜᴄᴛᴜʀᴇ [29] for (*a*) all seven pre-defined populations, where two main clusters (*K* = 2)—inland and coastal—were identified. (*b*) Cluster analysis for only inland populations (*K* = 4) and (*c*) coastal populations (*K* = 3).

**Table 3.** Analysis of molecular variance (AMOVA) for baobab (*A. digitata*) populations in southeastern Kenya show variation among and within groups/populations. (AMOVA was assessed for all seven populations, for the coastal and inland group, for coastal and inland individuals, and for inland (four) and for coastal populations (three) separately. d.f., degrees of freedom; *p* < 0.001, 10 000 permutations.)

|  | d.f. | sum of squares | variance components | percentage of variation |
|---|---|---|---|---|
| among all seven populations | 6 | 267.48 | 1.34 | 9.40 |
| within all seven populations |  | 2358.96 | 12.96 | 90.60 |
| among inland and costal group | 1 | 179.29 | 3.71 | 22.07 |
| within inland and costal group |  | 2447.15 | 13.09 | 77.93 |
| among four inland populations | 3 | 56.01 | 0.15 | 1.12 |
| within four inland populations |  | 2106.34 | 13.25 | 98.88 |
| among three coastal populations | 2 | 32.18 | 0.61 | 5.23 |
| within three coastal populations |  | 252.63 | 10.98 | 94.77 |

# 4. Discussion

## 4.1. Genetic diversity and differentiation

Only a few studies have been published on the genetic diversity of *A. digitata* using microsatellites [2,6,13]. Genetic diversity represented by He and *H* was shown to be similar to Sudanese baobab populations [2]. Slightly higher values of *H* were observed in Malawian baobab populations [13]. In West Africa, AFLP markers were used to estimate genetic diversity and genetic structure of *A. digitata*. Although not directly comparable to our findings, genetic diversity was also high

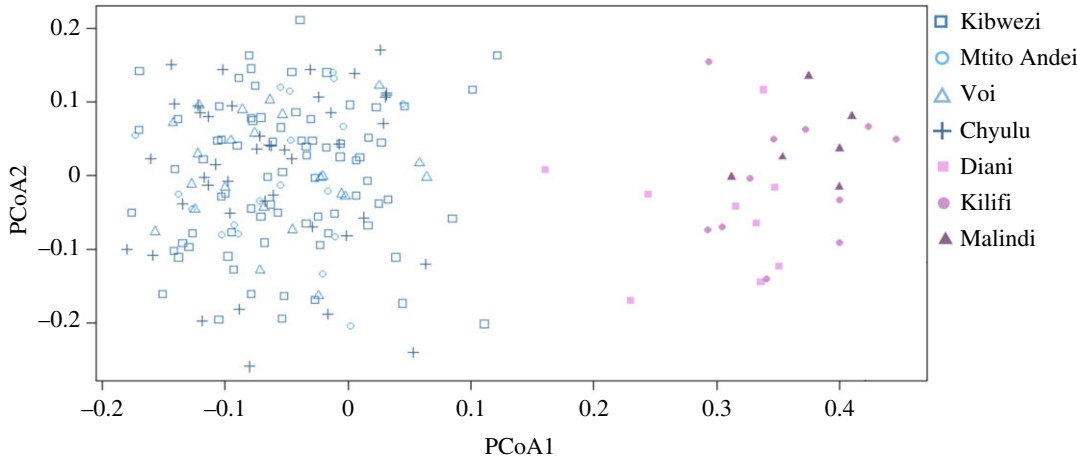

**Figure 3.** Principal coordinate analysis (PCoA, Bruvo distance) of the seven analysed baobab populations in southeastern Kenya based on a dataset of nine microsatellite loci. PCoA axes 1 and 2 represent 15.1% and 6.1% variation, respectively. On the left side, individuals (highlighted in blue colours) from inland populations grouped together: Kibwezi, Mtito Andei, Voi and Chyulu. On the right side, there is a group of coastal individuals (in violet) from Diani, Kilifi and Malindi populations.

[5,11,12]. Baobab populations across the African continent seem to possess high genetic diversity. Such populations are generally considered to be healthy, with greater resistance to pest and disease, and lower susceptibility to environmental changes [36]. Genetic diversity and population structure of a plant species are directly influenced by its reproductive system and gene flow [37]. Baobab is a self-incompatible species [38] possessing a reproductive mechanism common for most angiosperm plants which help to maintain and increase diversity within populations as cited in [37]. The major baobab pollinators are fruit bats [39], and seed dispersal is commonly mediated by humans and many animals (e.g. elephants, baboons, chimpanzees, eland) [17]. The movement of pollinators and dispersal agents has an important role in the current genetic diversity of baobabs. Furthermore, the level of diversity is influenced by the baobab's tetraploid nature. Polyploids are generally known to have higher levels of genetic diversity compared to diploid species, owing to a larger total number of chromosome copies [10]. Thus, polyploid species may result in broader ecological amplitudes [40,41].

Genetic differentiation ($\rho$) was low within both inland and coastal regions which indicates sufficient gene flow within both regions. Similarly, in Sudan, low genetic differentiation was found among populations in the Nuba Mountains [2]. Nevertheless, substantially higher $\rho$ values were observed among both regions in Kenya. This fact suggests rather limited gene flow between inland and coastal populations. Interestingly, patterns of genetic differentiation are not correlated to geographical distances, as the geographical distance between Voi (inland, but closest to coast) and Kilifi (coast, but closest to inland) was almost similar to those between the two inland populations Kibwezi and Voi (figure 1), but reported $\rho$ values differ strongly (0.320 and 0.022, respectively). A similar observation was made in Malawi where the closest geographical populations (56 km) were clustered separately, whereas populations 204 km apart were clustered together [13]. This implies that geographical distance is not the main factor that shapes patterns of baobab diversity, which is not uncommon in a semi-domesticated species [42].

In the present study, most genetic variation (greater than 90%) was found within the seven surveyed populations and 9.4% of the variation was observed among populations based on AMOVA analysis. Results are in line with the findings of Wiehle *et al.* [2], who also reported low variation among (7.1%) and high within-location variation (92.9%) in Sudan. However, AMOVA was also performed among and within inland and the coastal regions only, where substantially higher values between two regions were found (22.07%). These results support the fact that there is a limited gene flow between coastal and inland regions.

## 4.2. Population structure

Bayesian clustering and PCoA separated baobabs into two groups according to their geographical origin, i.e. to inland and coastal clusters. One of the factors to explain these findings may be elevation; inland

populations are found in higher altitudes (greater than 600 m.a.s.l.) while coastal populations are found close to sea level (less than 60 m.a.s.l.), with substantially different climate conditions. Therefore, climate and elevation may explain the current population structure. Moreover, baobabs have different phenologies in the two regions. Based on personal observations, inland trees bore fruit earlier than those at the coast. This is most likely caused by different flowering times, and therefore, the gene flow between both regions is very limited. The main baobab pollinators, fruit bats [39], may play a significant role in shaping genetic structure [37]. The fruit bat *Rousettus aegyptiacus* was reported to pollinate baobab flowers at the coastal region in Kenya [43], however, no information is available for the inland region. Bats may contribute to long-distance pollen dispersal and ensure the genetic connectivity [44], but *R. aegyptiacus* might not be responsible for the pollination of baobab in inland regions and therefore not supporting gene flow between coastal and inland baobab populations. However, the behaviour of bats is largely unknown, only a few studies have been carried out (as cited in [4,39]) and more research with a focus on pollination of baobabs should be performed to truly understand the pollinator's behaviour and its effects on baobab populations.

The clear differences between inland and coastal clusters as shown in the present study were observed in a similar way in some other studies. In Benin, for example, coastal populations (in the Guinean climatic zone) were also clearly divided from inland populations [5]. A study by Bell *et al.* [6] also described strong differentiations between inland and coastal clusters for baobabs sampled in East and southeast African regions (Mozambique, Tanzania, Kenya and South Africa). The baobab is a typical savannah tree, and coastal climatic conditions, where the climate is more humid, may be limiting for the occurrence of the tree. Thus, Assogbadjo *et al.* [5] explained Benin's baobab distribution by climatic changes (dry era) in the late Holocene about 3700 years BP in the coastal area. These climatic conditions allowed the baobab to reach the coast and to grow there. According to them, later, the baobabs may have adjusted to the more humid conditions, which can be found there nowadays. Therefore, a similar change of climatic condition may also have occurred in Kenya and other coastal areas of East Africa. Apart from the climate, the different edaphic conditions of the coastal areas may have acted as a factor of selection pressure on African baobab populations [45]. As the coastal soils are often formed on calcareous sedimentary rocks, the baobabs may have adapted to the alkalinity and lack of nutrients typical for these sites, as edaphic adaptation is a known driver of evolution of ecotypes and speciation in plants [46,47]. Another possibility is a more recent introduction of baobabs to the coastal region by humans. Ancient Africans who belonged to farming, nomadic and fishing communities were familiar with the numerous uses of baobabs, and they carried the lightweight and nutritious baobab fruits when travelling overland [6]. Such journeys might have introduced the baobabs to the coastal regions where the trees adjusted to the new environment without being able to exchange pollen with inland individuals and over time, they evolved to a coastal ecotype, because both factors influencing evolution, i.e. selection pressure and a barrier to gene flow, are present [48].

However, the reason for such genetic differentiation remains unrevealed. Most probably, different population histories associated with climate change or even human intervention could have caused the current genetic differentiation. Therefore, nowadays baobabs from both regions are not capable of gene exchange owing to different phenology and flowering time. The fact that those findings were also observed in other parts of Africa e.g. Benin, Mozambique, Tanzania, Kenya [5,6] means that further studies should look if such genetic differentiation also reflects some morphological changes within the trees. If yes, it could reveal two distinct baobab ecotypes (coastal and inland), with a special adaptation to climatic and environmental conditions of the two locations.

## 5. Conclusion

This study has revealed high genetic diversity in all studied baobab populations and strong differentiation between inland and coastal populations. The differentiation cannot be easily explained but may have been caused by different population histories supported by particular climatic conditions, which prevent genetic exchange between the two study regions. Thus, gene flow mostly occurs within populations in both regions, a fact which is also important for conservation and management issues. The results of this study contribute to the general knowledge of baobab trees in East Africa and may help to further conservation and domestication programmes. Selection of baobab elite trees for domestication should be done separately in the two study regions to maintain the genetic diversity of the species. We also recommend more intensive sampling and analysis of the

genetic and morphological characterization of coastal and inland baobab populations across Africa to assess if such differentiation could also be found in other parts of the continent. This would be helpful in revealing new facts about phylogeography, genetics and morphology of this species.

Data accessibility. The datasets supporting this article have been uploaded as part of the electronic supplementary material, files SM1 (SM_1 genetic data) and electronic supplementary material, SM2 (SM_2 structure results).

Authors' contributions. A.C. carried out the fieldwork in Kenya, the laboratory work, data analysis and prepared the main draft of the manuscript; K.K. initiated the research project and framed the overall research objectives; M.K., B.M., K.P. and J.Š. carried out some analyses of the data and helped with preparation of the manuscript; P.V.D. reviewed the final manuscript and did English corrections; B.L. supervised the study and led the write-up of the paper. All authors gave final approval for publication.

Competing interests. We have no competing interests.

Funding. The authors wish to acknowledge the funding provided by the European Commission (EC) and the International Fund for Agricultural Development (IFAD) for the Fruiting Africa Project led by the World Agroforestry Centre (ICRAF). This study was also funded by the Internal Grant Agency of CULS Prague (CIGA no. 20185004, IGA no. 20195005) and the Foundation Nadace Nadání Josefa, Marie a Zdeňky Hlávkových, Czech Republic (Foundation Grant).

Acknowledgements. We would like to thank a number of Kenyan government stakeholders, i.e. Kenya Forestry Research Institute (KEFRI), Kenya Forest Service (KFS) and Ministry of Agriculture (MoA), Kenya Wildlife Service (KWS), and the local farmers and key informants for their help and guidance during baobab tree sampling across southeastern Kenya. Thanks also to Charles Waruhiu, Joseph Awuol and George Niakundi who helped during the sampling.

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
