## [Reviewer comments · Royal Society Open Science]

Review History

RSOS-190854.R0 (Original submission)

Review form: Reviewer 1

Is the manuscript scientifically sound in its present form?

Yes

Are the interpretations and conclusions justified by the results?

Yes

Is the language acceptable?

Yes

Do you have any ethical concerns with this paper?

No

Have you any concerns about statistical analyses in this paper?

No

Recommendation?

Major revision is needed (please make suggestions in comments)

Comments to the Author(s)

I read your manuscript. It reads well, but is partly lacking thorough information, which I tried to compensate through the review (Appendix A). terminology needs to be more strictly applied "population" and "accession" for instance. Moreover, parts of the paper contain information that is well known and does not need to be written (over and over again) in such kind of articles. The discussion part needs to be more elaborated and adapted to the specific environmental conditions and history of the study region!

Review form: Reviewer 2 (Achille Assogbadjo)

Is the manuscript scientifically sound in its present form?

Yes

Are the interpretations and conclusions justified by the results?

No

Is the language acceptable?

No

Do you have any ethical concerns with this paper?

No

Have you any concerns about statistical analyses in this paper?

No

Recommendation?

Major revision is needed (please make suggestions in comments)

Comments to the Author(s)

Overall, the paper can be published if all issues pointed out are carefully revised (see Appendix B).

Decision letter (RSOS-190854.R0)

22-Jul-2019

Dear Dr Lojka,

The editors assigned to your paper ("Genetic diversity and structure of baobab (*Adansonia digitata* L.) in south-eastern Kenya") have now received comments from reviewers.

While both reviewers are positive about publication of your paper, they both raise very substantive points about the manuscript as it stands. We would like you to revise your paper in accordance with the referee suggestions which can be found below (not including confidential

reports to the Editor). Two PDFs have been included with this message, while a third is being sent separately by Royal Society Open Science, as the file size exceeds the limit imposed by ManuscriptCentral ScholarOne. Please ensure this file is circulated among your colleagues, too. Please note this decision does not guarantee eventual acceptance.

Please submit a copy of your revised paper before 14-Aug-2019. Please note that the revision deadline will expire at 00.00am on this date. If we do not hear from you within this time then it will be assumed that the paper has been withdrawn. In exceptional circumstances, extensions may be possible if agreed with the Editorial Office in advance. We do not allow multiple rounds of revision so we urge you to make every effort to fully address all of the comments at this stage. If deemed necessary by the Editors, your manuscript will be sent back to one or more of the original reviewers for assessment. If the original reviewers are not available, we may invite new reviewers.

- Data accessibility

<http://datadryad.org/submit?journalID=RSOS&manu=RSOS-190854>

- Competing interests

- Authors' contributions

- Acknowledgements

- Funding statement

Kind regards,
Andrew Dunn
Senior Publishing Editor
Royal Society Open Science
openscience@royalsociety.org

on behalf of Dr James Locke (Associate Editor) and Steve Brown (Subject Editor)
openscience@royalsociety.org

Associate Editor's comments (Dr James Locke):

Associate Editor: 1

Comments to the Author:

Please follow the suggestions outlined by the two reviewers in your revision.

Comments to Author:

Reviewers' Comments to Author:

Reviewer: 1

Comments to the Author(s)

I read your manuscript. It reads well, but is partly lacking thorough information, which I tried to compensate through the review (txt file). terminology needs to be more strictly applied

"population" and "accession" for instance. Moreover, parts of the paper contain information that is well known and does not need to be written (over and over again) in such kind of articles. The discussion part needs to be more elaborated and adapted to the specific environmental conditions and history of the study region!

Reviewer: 2

Comments to the Author(s)

Overall, the paper can be published if all issues pointed out are carefully revised.

Author's Response to Decision Letter for (RSOS-190854.R0)

See Appendices C & D.

Decision letter (RSOS-190854.R1)

21-Aug-2019

Dear Dr Lojka,

I am pleased to inform you that your manuscript entitled "Genetic diversity and structure of baobab (*Adansonia digitata* L.) in south-eastern Kenya" is now accepted for publication in Royal Society Open Science.

on behalf of Dr James Locke (Associate Editor) and Steve Brown (Subject Editor)
openscience@royalsociety.org

Associate Editor Comments to Author (Dr James Locke):

The authors have revised the manuscript in according to the comments from the referees and the paper is now acceptable for publication.

Appendix A

Page 4

Line 28: "...to be a semi-domesticated..." or "...to be semi-domesticated [-species]"

Page 5

Line 24 & 29: decide for one "-" or "to" to give ranges of values (in this case precipitation)

Line 35-54: not sure if the description of general botanic knowledge about this species is really need. I kindly request to either bring in general observations from the field in Kenya that or to delete it completely. The last sentence's content, however, is important for the context of the article!

Page 6

Line 3: 189 is unequal to 253 (43 + 64 + 146)!

Line 6: "measured" what? Have morphologies been measured?

Line 6: "146" beginn a sentence only with written out numbers or restructure the sentence

Line 7-13: wording is odd

Line 26: "a CTAB" ="the CTAB"

Line 26ff: manufacturers should be given including city name when named first; Carolina Biosystems has no city/country

Page 7

Line 12: Fis is no diversity measure, therefore kindly put into the section "differentiation"

Line 16: Why has only Fis calculated? First, it is not justified, second, Fst, Gst or other diferentiation measures are more often used in population genetic studies, which thenn also allows a better comparison.

Line 18: "...using." sentence incomplete

Line 24 ff.: Why a binary matrix was formed? It is not wrong, but a reasoning is missing!

Line 26: Why Arlequin was approached, when AMOVA can be also calculated with GenAleEx?

Line 32: "PCoA (Principal Coordinate Analysis)" = "Principal Coordinate Analysis (PCoA)"

Line 47: Why different cluster adjustments for different sample regions? I think the answer is written on Page 7, Line 49-51. Thus, kindly move the reasoning from these lines after these lines 47ff!

Page 8

Line 9: "3.9 to 6.0" or "4 to 6"

Line 13: "was high as expected in tetraploid individuals (0.907)." interprets already the finding; hence it is part of the discussion

Line 14: Please explain how negative Fis values relate to your results!

Line 14-15: "as expected" implies interpretation of results; hence discussion

Line 20: "Genetic differentiation (?) was computed for the seven populations and compared with the geographical distances

(Table 2)." reads more than Materials and Methods part. Please correct

Line 36: "Analysis of molecular variance" is written out her, but not in the Materials Methods section Page 7, Line 26. Please only write AMOVA here and add full term on Page 7

Line 45: "...two clusters are informative." is poorly written. Kindly provide the full truth: Delta K (highest), SD (likely lowest, at least low)

Page 9

Line 13: "He (gene diversity corrected for sample size)": if so, this would be a note for the Materials Methods section. Second: There are well defined measures corrected for sample size, which is Band richness and allelic richness. Both are likely the same but cannot be calculated based on the softwares shown. Kindly correct this by giving the appropriate software (BTW: Page 6, missing software!) or delete the statement!

Line 24: "...clearly been influenced by people." sounds as people would modify individuals. I guess the kind of admixture of plant preferred/superior plant material concentrated at a certain area is meant here. Please reword!

Line 30-34: kindly split this sentence into two and add "...may result in broader ecological amplitudes [35,36], as shown in our study."

Page 10

Line 11: "...over- or underestimating population genetic structure." = "over- or underestimation of population genetic structure."

Line 16: "as most informative" = "as most likely"

Line 18: "...one the factors..." = "...one of the factors..."

Line 22: "phenological stages"? Where have they been presented? For what purpose?

Line 24-41: this entire section is not logically and argumentatively well written. The fact of different flower phenologies and times was even not stated in the introduction. The fact of different flowering times by region can basically be narrowed down to one sentence. One more sentence regarding the potential vectors and the need for pollination experiments - THAT'S IT!

Line 53: Space between units: "24°C" = "24 °C"

Line 55: "...trees, and.." = "...tree and.."

Line 57: "of the tree." = "of this species."

Page 11

Line 6-7: authors should also consider the soil structure along coast line(s?), definitely in Kenya, but maybe also in Ghana, Benin, ... Coast lines in Kenya are highly calcareous/permeable for water and may "imitate" savannah conditions, even though these populations receive far more precipitation (diluting effect on salt) than the inland populations. I personally do not see an adoption of baobab along the coast line (as assumed by Assogbadjo) but a selection of individuals towards saline conditions in due time. Moreover, what about effects of "recently" (300, 400, maybe 600 years back?) introduced material by for instance arabic people during times of colonizing East African coast lines? This aspect is not yet elaborated. Hints may derive from the study of Bell et al.!

Line 13: "The most prop...." = "Most properly,"

Line 7ff: "A study by Bell..." needs to be moved up when talking about clustering of different populations.

Line 14 ff: kindly reword the last two sentences; English and Logic is not well done.

Line 12: Conclusion"s"

References:

- check reference. Species names are not in italics

Figure 1:

- Move the overview map to the left, the arrows do not need to cross half of the figure

Figure 2

- kindly use same dimension of Figures 2B and 2C as for 2A. 2B and 2C are then of course not alligned-
- creat ONE x-axis legend only including sampling locations (larger Font!!!) and below 1x "Inland" and 1x "Coast"
- in the caption you talk about "accessions", on Figure 1 you talk about "populations"! Kindly decide for one term, check also the entire text (and figure, table sections) for the different names and correct for one or the other. I suggest to use accession (or location), since "populuation" was so far not really well defined!
- lines between accessions should be white in order to clearly differentiated

Figure 3:

- delete figure title
- Change "pca[,1]" etc. to "PCoA 1"... or "Coordinate 1".... Also change in caption!
- legend is far too small! Moreover, NO need to abbreviate the locations here as there is suffcient white space!

Table 1:

- Observed Heterozygosity was not mentioned in Materials and Methods!
- Caption should start with "average location and genetic parameters..." or so
- Code KIBin,... etc. is in my opinion not needed! Also not for Table 2 as names are not too long and abbreviations are ratherwv confusing. Delete then as well "[in= inland population, co= coastal population]"
- What means the parameter AR?
- as sample sizes strongly differ across locations, I strongly recommend to use only size-corrected measures as allelic richness etc...!
- apart of the caluclated parameters, private/unique alleles may be more important to explain accession-wise genetic differences and "value" of these accessions.
- as Fis was also presented you have to add the term/statement "...and differentiation parameters..." in the caption, as Fis (and other differentiation measures) are NOT diversity measures!!! I would even suggest to add this (and Fst) to Table 2 as a sub-table!!!

Appendix B

Review report for Genetic diversity and structure of baobab (*Adansonia digitata* L.) in south-eastern Kenya

The study tackled genetic diversity of baobab populations in two geographical regions of Kenya. In general, high within-pop and low among-pop variations were observed. Although, the study used a sound methodological approach to unravel Microsats-based diversity, the writing style still a bit poor.

The introduction merely gather facts without pointing out problems that the study aims to solve. No study justification was provided except the usual lack of study.

The methodology is clear enough, though some precisions can be added (see in manuscript file) The reporting of results is okay, some few improvement that can be made are suggested in the manuscript file.

The discussion section is too shallow, with some elements of literature review thrown as elements of discussion. Comparisons with previous studies are not accurate and deviations are not explained nor commented on.

Crosscheck references for consistency and confirm they follow the format required by the journal.

Captions of figure can be improved

Overall, the paper can be published if all issues pointed out are carefully revised.

Appendix C

Review report for Genetic diversity and structure of baobab (*Adansonia digitata* L.) in south-eastern Kenya

The study tackled genetic diversity of baobab populations in two geographical regions of Kenya. In general, high within-pop and low among-pop variations were observed. Although, the study used a sound methodological approach to unravel Microsats-based diversity, the writing style still a bit poor.

The introduction merely gather facts without pointing out problems that the study aims to solve. No study justification was provided except the usual lack of study.

We modified the background accordingly, more justification of the study was added.

The methodology is clear enough, though some precisions can be added (see in manuscript file) The reporting of results is okay, some few improvement that can be made are suggested in the manuscript file. Above mentioned precisions were added, and some improvements or justifications were done.

The discussion section is too shallow, with some elements of literature review thrown as elements of discussion. Comparisons with previous studies are not accurate and deviations are not explained nor commented on. The entire discussion was changed extensively. We arranged it more logically, to provide more in-depth discussion of all results, more comments to the discussion through the following text.

Crosscheck references for consistency and confirm they follow the format required by the journal. This was done.

Captions of figure can be improved. Captions were improved.

Overall, the paper can be published if all issues pointed out are carefully revised.

The point-by-point responses are in following pages, either marked in red or as direct responses to the comments in pdf.

The responses to the reviewer's comments are marked red

Page 4

Line 28: "...to be a semi-domesticated..." or "...to be semi-domesticated [-species]"
This has been corrected

Page 5

Line 24 & 29: decide for one "-" or "to" to give ranges of values (in this case precipitation)

This has been corrected

Line 35-54: not sure if the description of general botanic knowledge about this species is really need. I kindly request to either bring in general observations from the field in Kenya that or to delete it completely. The last sentence's content, however, is important for the context of the article! The general botanic description was removed

Page 6

Line 3: 189 is unequal to 253 (43 + 64 + 146)! 64 is number of farms not sampled trees. The total number of sampled trees is 189

Line 6: "measured" what? Have morphologies been measured? The word "measured" was removed

Line 6: "146" beginn a sentence only with written out numbers or restructure the sentence. This has been corrected

Line 7-13: wording is odd Wording was changed

Line 26: "a CTAB" ="the CTAB". This has been corrected

Line 26ff: manufacturers should be given including city name when named first; Carolina Biosystems has no city/country. This was added

Page 7

Line 12: Fis is no diversity measure, therefore kindly put into the section "differentiation". Fis is inbreeding coefficient, and is commonly added to a summary genetic statistics. However, the caption of the table 1 was modified accordingly.

Line 16: Why has only Fis calculated? First, it is not justified, second, Fst, Gst or other differentiation measures are more often used in population genetic studies, which thenn also allowws a better comparison. Population differentiation was computed. We used ρ statistic Meirmans et al. [10], as they explain that ρ statistic is the most relevant alternative to F_{ST} for polyploid species. This is mentioned in methodology.

Line 18: "...using." sentence incomplete This has been corrected

Line 24 ff.: Why a binary matrix was formed? It is not wrong, but a reasoning is missing! It was created in order to compute AMOVA and Nei's genetic diversity, which allowed us comparison with other studies.

Line 26: Why Arlequin was approached, when AMOVA can be also calculated with GenAleEx? The issues of polyploids

Line 32: "PCoA (Principal Coordinate Analysis)" = "Principal Coordinate Analysis (PCoA)" This has been corrected

Line 47: Why different cluster adjustments for different sample regions? I think the answer is written on Page 7, Line 49-51. Thus, kindly move the reasoning from these lines after these lines 47ff! This has been corrected. The justification os different cluster adjustments was moved to the methodology: "Janes et al. [30] reported that ΔK frequently identifies $K = 2$ even if more subpopulations are present, therefore it

may cause an over- or underestimating population genetic structure. To prevent this, the division of samples was done to reveal if there is more genetic structure within each of the two regions."

Page 8

Line 9: "3.9 to 6.0" or "4 to 6" **This has been corrected**

Line 13: "was high as expected in tetraploid individuals (0.907)." interprets already the finding; hence it is part of the discussion **This has been corrected**

Line 14: Please explain how negative F_{is} values relate to your results! **Explanation was added.**

Line 14-15: "as expected" implies interpretation of results; hence discussion **This has been corrected**

Line 20: "Genetic differentiation (?) was computed for the seven populations and compared with the geographical distances (Table 2)." reads more than Materials and Methods part. Please correct

Line 36: "Analysis of molecular variance" is written out here, but not in the Materials Methods section Page 7, Line 26. Please only write AMOVA here and add full term on Page 7 **This has been corrected**

Line 45: "...two clusters are informative." is poorly written. Kindly provide the full truth: Delta K (highest), SD (likely lowest, at least low). **Rewriting of the sentence was done. The Evanno statistics are provided in supplementary material.**

Page 9

Line 13: "He (gene diversity corrected for sample size)": if so, this would be a note for the Materials Methods section. – **This has been moved to MM section.** Second: There are well defined measures corrected for sample size, which is Band richness and allelic richness. Both are likely the same but cannot be calculated based on the softwares shown. Kindly correct this by giving the appropriate software (BTW: Page 6, missing software!) or delete the statement! **Not aware of software, which could calculate these parameter for polyploids**

Line 24: "...clearly been influenced by people." sounds as people would modify individuals. I guess the kind of admixture of plant preferred/superior plant material concentrated at a certain area is meant here. Please reword! **This has been reworded.**

Line 30-34: kindly split this sentence into two and add "...may result in broader ecological amplitudes [35,36], as shown in our study." **This has been corrected**

Page 10

Line 11: "...over- or underestimating population genetic structure." = "over- or underestimation of population genetic structure." **This has been corrected**

Line 16: "as most informative" = "as most likely" **This has been corrected**

Line 18: "...one the factors..." = "...one of the factors..." **This has been corrected**

Line 22: "phenological stages"? Where have they been presented? For what purpose? **This was stated in the discussion in order to explain the two different genetic clusters (inland and coastal regions)**

Line 24-41: this entire section is not logically and argumentatively well written. The fact of different flower phenologies and times was even not stated in the introduction. The fact of different flowering times by region can basically be narrowed down to one sentence. One more sentence regarding the potential vectors and the need for isolation experiments - THAT'S IT! **this entire paragraph was changed extensively.**

We arranged it more logically, to provide more in depth discussion of genetic structure results

Line 53: Space between units: "24□C" = "24 □C" This has been corrected

Line 55: "..trees, and.." = "..tree and.." This has been corrected

Line 57: "of the tree." = "of this species." This has been corrected

Page 11

Line 6-7: authors should also consider the soil structure along coast line(s?), definitely in Kenya, but maybe also in Ghana, Benin,... Coast lines in Kenya are highly calcereous/permeable for water and may "immitate" savannah conditions, even though these population receive far more precipitation (diluting effect on salt) than the inland populations. I personally do not see an adoption of baobab along the coast line (as assumed by Assogbadjo) but a selection of individuals towards saline conditions in due time. **These statement was consider and add to the discussion part.** Moreover, what about effects of "recently" (300, 400, maybe 600 years back?) introduced material by for instance arabic people during times of colonizing East African coast lines? This aspect is not yet elaborated. Hints my derive from the study of Bell et al.! **The part about introducing baobabs to the coastline by people was added.**

Line 13: "The most prop...." = "Most properly," This has been corrected

Line 7ff: "A study by Bell..." needs to be moved up when talking about clustering of different populations. **This has been moved up**

Line 14 ff: kindly reword the last two sentences; English and Logic is not well done.

Line 12: Conclusion"s" **ok**

References:

- check reference. Species names are not in italics. **Only first reference was in italic, because it is a book title. However, we checked all the references**

Figure 1:

- Move the overview map to the left, the arrows do not need to cross half of the figure. **This figure was modified accordingly.**

Figure 2

- kindly use same dimension of Figures 2B and 2C as for 2A. 2B and 2C are then of course not alligned-

- creat ONE x-axis legend only including sampling locations (larger Font!!!) and below 1x "Inland" and 1x "Coast"

- in the caption you talk about "accessions", on Figure 1 you talk about "populations"! Kindly decide for one term, check also the entire text (and figure, table sections) for the different names and correct for one or the other. I suggest to use accession (or location), since "populuation" was so far not really well defined!

- lines between accessions should be white in order to clearly differentiated

This figure and its caption was modified as suggested.

Figure 3:

- delete figure title

- Change "pca[,1]" etc. to "PCoA 1"... or "Coordinate 1".... Also change in caption!

- legend is far too small! Moreover, NO need to abbreviate the locations here as there is sufficient white space! **This figure and its caption was modified as suggested.**

Table 1: **This table was modified:**

- Observed Heterozygosity was not mentioned in Materials and Methods! **Added to MM**
- Caption should start with "average location and genetic parameters..." **In caption was modified "genetic parameters" instead of "genetic diversities"**
- Code KIBin,... etc. is in my opinion not needed! Also not for Table 2 as names are not too long and abbreviations are rather confusing. Delete then as well "[in= inland population, co= coastal population]" **This has been done**
- What means the parameter AR? **A mistake**
- as sample sizes strongly differ across locations, I strongly recommend to use only sizecorrected measures as allelic richness etc...!
- apart of the calculated parameters, private/unique alleles may be more important to explain accession-wise genetic differences and "value" of these accessions. **However, polyploid species rise complications in missing dosage information... if these parameters would be calculated; the results may be biased...**
- as Fis was also presented you have to add the term/statement "...and differentiation parameters..." in the caption, as Fis (and other differentiation measures) are NOT diversity measures!!! I would even suggest to add this (and Fst) to Table 2 as a sub-table!!! **Fis commonly added to a summary genetic statistics**

Appendix D

Page 5

The introduction is unnecessarily long, pointing out general information without tackling a real problem that led to the undertake of this study. No study justification is provided. Although the choice of Microsats has been justified, it is not justified why the need to assess genetic diversity of the populations. This cannot be justified only by the lack of study, maybe the lack of study is a result of no need for it
We modified the background accordingly; more justification of the study was added.

Page 6

You need to specify whether tree sampling was done randomly. Also you should define the population meaning, what did you consider as population?
The information about the sampling design was improved.

Page 7

Line 8: This sentence is incomplete. **This has been corrected.**

Page 8

Line 26: Add statistics here. **This has been added.**

Line 43: provide a range here. **This has been added.**

Page 9

Line 5-11: this is better placed in the methodology. **It was moved to the methodology.**

Line 18-34: explain the why of such a deviation and tell us what could be the implication of such diff. what do you mean? and how does this compare to your findings? In this study you have no evidence for saying this. hanging sentences with no link to your study objective. **This entire paragraph was changed extensively. We arranged it more logically, to provide more in depth discussion of genetic diversity results.**

Line 51: so what? **The paragraph was more logically ordered.**

Line 55-56: what are you trying to say here? **This sentence was deleted.**

Page 10

Line 3: not clear. **The sentence was improved.**

Line 9-15: compare and discuss the results, no need to recall here what you did and why. this should have been said in the MM section. **We moved this part in MM.**

Line 16-17 this element could be in the result, not here as element of discussion. **This was done.**

Line 24-25: is there any gradient that could be clearly observed? what did previous studies report on altitude/climatic conditions and variation in population structure of baobab? **There is no clear gradient being observed, the climatic conditions from previous studies are included in discussion.**

Line 30: rainy or dry season there? **Sentence was changed.**

Line 38: you can elaborate more on this. **It was done.**

Line 40-41: always relate things to your study objectives. **We consider that identifying factors that influence gene flow relates to the objectives of our study.**

Line 45-52: here you are trying to review literature which should not be the case in a discussion. you should always link to your results and study objective. **This section was shortened and rewrite to better explain the similarities between our results and results from the Assogbadjo et al [8]**

Line 52-53: how does this provides insights to your study objective? **This sentence was removed.**

Page 12

References: Check and confirm that all citations are listed here and all reference have been cited above. Also, check for consistency in regards to the journal's requirement. **All references were checked.**

Page 16-18

Figure 1: better if you can get this justified rather than keeping it centered. **The position of the caption was generated by the submission system, however this figure was improved.**

Figure 2 See comment above. **Same response as for figure 1.**

Figure 3 See above. **Same response as for figure 1.**

Page 19

Table 1: How can mean of this be higher than all observed values? **This row was generated by the SPAGEDI software, and shows values for all populations instead of "mean". This issue was fixed.**